# INTERPRETING EQUIVARIANT REPRESENTATIONS

## ABSTRACT

Latent representations are used extensively for downstream tasks, such as visualization, interpolation or feature extraction of deep learning models. Invariant and equivariant neural networks are powerful and well-established models for enforcing inductive biases. In this paper, we demonstrate that the inductive bias imposed on the by an equivariant model must also be taken into account when using latent representations. We show how not accounting for the inductive biases leads to decreased performance on downstream tasks, and vice versa, how accounting for inductive biases can be done effectively by using an invariant projection of the latent representations. We propose principles for how to choose such a projection, and show the impact of using these principles in two common examples: First, we study a permutation equivariant variational auto-encoder trained for molecule graph generation; here we show that invariant projections can be designed that incur no loss of information in the resulting invariant representation. Next, we study a rotation-equivariant representation used for image classification. Here, we illustrate how random invariant projections can be used to obtain an invariant representation with a high degree of retained information. In both cases, the analysis of invariant latent representations proves superior to their equivariant counterparts. Finally, we illustrate that the phenomena documented here for equivariant neural networks have counterparts in standard neural networks where invariance is encouraged via augmentation. Thus, while these ambiguities may be known by experienced developers of equivariant models, we make both the knowledge as well as effective tools to handle the ambiguities available to the broader community.

## 1 INTRODUCTION

Latent representations are used extensively in the interpretation and design of deep learning models. The latent spaces of VAEs (Kingma et al., 2019) and masked autoencoders (He et al., 2022), learned from great amounts of unlabeled data in an unsupervised or self supervised manner, are prominent examples of powerful latent representations. These representations are used for e.g. molecular-, chemical- or protein discovery (Detlefsen et al., 2022); image generation, synthesis (Goodfellow et al., 2020) and segmentation (Kirillov et al., 2023); interpretability using visualization tools like T-SNE (Van der Maaten & Hinton, 2008), UMAP (McInnes et al., 2018) or PCA on latent embeddings; or counterfactual explainability (Papernot & McDaniel, 2018). It is natural to ask for such autoencoders – we use VAEs as a running example – to be equivariant. However, analysing equivariant latent features is no trivial task, and may yield misguided conclusions if done incorrectly.

$SO(2)$ Invariant Classifier

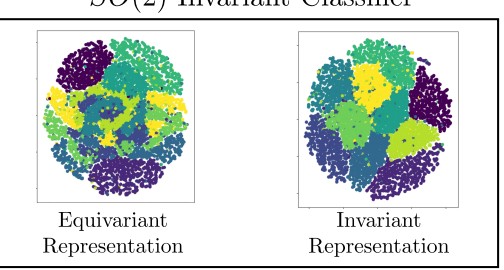

Equivariant Representation    Invariant Representation

Figure 1: Visualizing rotated MNIST images via their equivariant representation (left) hides structure that becomes apparent using an invariant representation of the same latent codes (right).

A geometric inductive bias can be imposed on deep learning models by requiring the model to be either invariant or equivariant to certain transformations of the data, usually represented by a group of symmetries (Cohen & Welling, 2016; Puny et al., 2021). In invariant models, the prediction is unchanged when group transformations act on the model's input. Mathematically, we formalize this

as follows: If $h\colon \mathcal{X} \to \mathcal{Y}$ is a neural network with a group $G$ acting on the input space $\mathcal{X}$, then $h$ is *G-invariant* if $h(g \cdot x) = h(x)$ for all $g \in G$. In equivariant models, the prediction has an analogous action of the group $G$ on the target space $\mathcal{Y}$, whose transformations are aligned with those acting on the inputs – as is the case, for instance, with equivariant autoencoders. Then $h$ is *G-equivariant* if $h(g \cdot x) = g \cdot h(x)$, aligning the predictions with their inputs according to the group action $G$.

Both invariant and equivariant networks are commonly designed with latent feature representations $\mathcal{Z}$, namely $h\colon \mathcal{X} \xrightarrow{f} \mathcal{Z} \xrightarrow{k} \mathcal{Y}$, where the latent space $\mathcal{Z}$ has an action of $G$, and the feature embedding $f$ is $G$-equivariant. We refer to the latent representation $\mathcal{Z}$ as an *equivariant representation*. Equivariant latent representations come with the caveat that any latent code $z = f(x) \in \mathcal{Z}$ will be equivalent to the – often different – representation $g \cdot z = g \cdot f(x) = f(g \cdot x)$ of the transformed, but equivalent, input $g \cdot x$. Thus, each input $x$ will be represented by the entire set $G \cdot z = \{g \cdot z | g \in G\}$ of latent vectors acted on by the group $G$. Different choices of $g$ – which are typically not made by the user but rather determined implicitly by the data collection – can lead to widely different latent embeddings $g \cdot z$, see Fig. 1. In this paper, we show how to derive invariant representations, such as the one shown in Fig. 1, that allow faithful visualization and utilization of equivariant latent codes.

**We contribute**   an empirical demonstration of how naïve interpretation of equivariant latent spaces can lead to incorrect conclusions about data and decreased performance of derived models. We mathematically explain how analysis of equivariant latent representations needs to take the group action into account. We provide explicit tools for doing so using invariant projections of latent spaces. We evaluate their effect via widely encountered group actions on two widely used model classes: 1) A permutation equivariant variational autoencoder (VAE) representing molecular graphs acted on by node permutations, where we obtain isometric invariant representations of the data, and 2) an equivariant representations of a rotation-invariant image classifier, where we showcase random invariant projections as a general and efficient tool for providing expressive invariant representations.

Finally, we show that the ambiguity of equivariant latent representation also extends to completely standard deep learning models, where invariance is encouraged via augmentation. An empirical example shows that even for such models, latent representations display behavior similar to that observed in equivariant latent spaces. Thus, while these ambiguities might be known and avoided by experienced developers of equivariant models, pointing them out and providing tools to handle them is of high relevance to all users that encode implicit biases via either equivariance or augmentation.

## 2   LATENT REPRESENTATIONS OF EQUIVARIANT MODELS ARE AMBIGUOUS

In this section, we aim to review the mathematical tools needed to discuss the equivariant representation illustrated in Fig. 1. As above, let $h$ be a model with a latent feature embedding $f$:

$$h\colon \mathcal{X} \xrightarrow{f} \mathcal{Z} \xrightarrow{k} \mathcal{Y}.$$

We consider $\mathcal{Z}$ to be a $G$-equivariant representation. That is, the feature embedding $f$ is $G$-equivariant and either: $k$ is $G$-equivariant (e.g. as in an autoencoder or a segmentation network) or $k$ is a $G$-invariant predictor (e.g. as in a classification or regression task).

Typically, $f$ and $k$ are themselves complex models consisting of many layers – in an autoencoder setup, $f$ would be the encoder and $k$ the decoder. This architecture prototypes a typical classification or regression network built with a geometric prior in mind, and has been suggested as a general blueprint for deep learning architectures (Bronstein et al., 2021). However, the interpretation and utilization of the equivariant latent representation $\mathcal{Z}$ is non-trivial, as any two points $x_1, x_2 \in \mathcal{X}$, which are equal up to some group element $g \in G$, i.e. $x_1 = g \cdot x_2$, will *not* necessarily map to the same latent representation – they may in fact map to latent representations $z_1 = f(x_1)$ and $z_2 = f(x_2) = g \cdot f(x_1)$ which are far apart in the latent space. If standard latent space analysis methods are applied out of the box on such equivariant representations, this can create problems with downstream analysis, which typically relies on similar data points having nearby latent representations.

The modelling-assumption that two elements $x_1$ and $x_2$ are equivalent, if there exists a group element $g$ transforming one into the other is encapsulated by the *quotient space* $\mathcal{X}/G$, which is the set consisting of all orbits $G \cdot x = [x] = \{g \cdot x | g \in G\}$ (Bredon, 1972). While quotient spaces

have been used extensively in classical approaches to statistics and machine learning with geometric priors, they often come with non-Euclidean structure and singularities, severely inhibiting the availability of tools for statistics, optimization, and learning (Feragen & Nye, 2020; Mardia & Dryden, 1989; Kolaczyk et al., 2020; Severn et al., 2022; Calissano et al., 2023). Equivariant models implicitly encode the structure of quotient spaces, while, from the viewpoint of implementation and optimization, Euclidean tools are available with all the computational advantages they may offer.

In fact, the modeling choice of picking an equivariant feature embedding $f$ implicitly induces a feature embedding $f'$ defined on the quotient spaces of the input space $\mathcal{X}$ and the latent space $\mathcal{Z}$:

$$
\begin{array}{ccc}
\mathcal{X} & \xrightarrow{\ f\ } & \mathcal{Z} \\
{\scriptstyle \pi}\downarrow & & \downarrow{\scriptstyle \pi} \\
\mathcal{X}/G & \dashrightarrow{\ f'\ } & \mathcal{Z}/G
\end{array}
$$

Where $\pi$ denotes the canonical projection. The induced feature embedding $f'$ can be defined by via representatives $x$ from each orbit. That is,

$$f'([x]) = [f(x)] \in \mathcal{Z}/G$$

for every orbit $[x] \in \mathcal{X}/G$. The quotient latent representation $\mathcal{Z}$ comes with a *quotient metric* defining distances between orbits:

$$d([z_1], [z_2]) = \min_{g_1, g_2 \in G} \|g_1 \cdot z_1 - g_2 \cdot z_2\| = \min_{g \in G} \|z_1 - g \cdot z_2\|.$$

This perfectly illustrates the problem: the equivariant latent representation $\mathcal{Z}$ is analyzed using arbitrary equivariant representatives $g_1 \cdot z_1$ and $g_2 \cdot z_2$ of the orbits $[z_1]$ and $[z_2]$, the relative distances between the latent codes $g_1 \cdot z_1$ and $g_2 \cdot z_2$, which would typically be used for downstream analysis, are not well defined, as they depend on the group elements $g_1$ and $g_2$ – and may be very different from the relative quotient distance $d([z_1], [z_2])$ within $\mathcal{Z}/G$. Thus, it is key that we enforce the modelling assumption before conducting any down-stream analysis of the data.

## 3 INVARIANT ANALYSIS OF EQUIVARIANT REPRESENTATIONS

As outlined in the previous sections, the relative distance between latent equivariant representations is ill-defined, due to the multiple equivalent representations of data. In this section, we discuss how invariant projections of the equivariant latent representation can be used to obtain invariant representations that give unambiguous latent embeddings.

As any equivariant function composed with an invariant function will be invariant, we can obtain invariant representations of our latent features by passing them through an invariant function. That is, we aim to find an invariant map $s\colon \mathcal{Z} \to \mathcal{Z}_s$, and use this to extract invariant latent features. An obvious invariant projection is the quotient projection $\pi\colon \mathcal{Z} \to \mathcal{Z}/G$ – but as discussed above, the quotient $\mathcal{Z}/G$ comes with severe limitations for analysis. Instead, we seek an invariant feature representation $s$ where the invariant latent space $\mathcal{Z}_s$ is Euclidean. Note that any such $s$ can necessarily be written as a composition of $\pi$ with another mapping $s'$:

**Proposition 1.** *Let $s : \mathcal{Z} \to \mathcal{Z}_s$ be an invariant, surjective function. Then $s$ induces a surjective function $s' : \mathcal{Z}/G \to \mathcal{Z}_s$ as*

$$s'([z]) = s(z) \qquad \forall [z] \in \mathcal{Z}/G. \tag{1}$$

*That is, the following diagram commutes:*

$$
\begin{array}{ccc}
\mathcal{Z} & \xrightarrow{\ s\ } & \mathcal{Z}_s \\
{\scriptstyle \pi}\downarrow & \nearrow{\scriptstyle s'} & \\
\mathcal{Z}/G & &
\end{array}
$$

As the map $s$ is chosen post hoc, a main challenge is how to choose $s$ and $\mathcal{Z}_s$ to retain signal from the quotient space $\mathcal{Z}/G$ while simplifying the analysis. Choosing $s(z) = 0$ is obviously invariant, but will destroy any signal in $\mathcal{Z}$.

Prop. 1 (proof in the appendix) highlights one of the problems of picking an arbitrary invariant map $s$, as we will in general only be guaranteed that $\mathcal{Z}_s$ is more coarse than $\mathcal{Z}/G$. We refer to elements of $\mathcal{Z}$ as equivariant representations and to elements of $\mathcal{Z}_s$ as invariant representations. In general, an invariant representation $s(z) \in \mathcal{Z}_s$ does not uniquely identify the orbits of an element $z \in \mathcal{Z}$. We argue that the quality of the invariant $s$ depends on whether it satisfies:

**Bijectivity:** If $s$ induces a bijection $s'$ an immediate consequence is that all orbits of elements in $\mathcal{Z}$ can be uniquely identified by the corresponding invariant representation.

**Isometry:** Equivariant/invariant models carry an inductive bias ensuring that elements that are equal up to the group action are handled similarly. Adhering to this modelling assumption, latent features of such elements should be close. When $s$ induces an isometric function $s'$, then this will hold exactly for the invariant representations, which is isometric to the quotient representation – i.e. elements in $\mathcal{Z}$ which are close up to the action of a group element will also be close in $\mathcal{Z}_s$.

In a special case visited below, where $\mathcal{Z}_s$ and $\mathcal{Z}/G$ are isometric and $\mathcal{Z}_s \subset \mathcal{Z}$, we will speak of $\mathcal{Z}_s$ as being an isometric *cross section*. In cases where such a cross section exists, we argue, that the equivariant representation of the latent features should be mapped onto the cross section prior to any subsequent analysis.

## 3.1  RETRIEVING AN ISOMETRIC CROSS SECTION: LATENT GRAPH REPRESENTATION

Next, we consider the special case of a permutation equivariant model $h : \mathcal{X} \xrightarrow{f} \mathcal{Z} \xrightarrow{k} \mathcal{X}$, where the group $G$ is the symmetric group $S_n$ of permutations on $n$ elements, and the latent representation is given by $\mathcal{Z} \subseteq \mathbb{R}^n$. For this particular case, we show, that we can choose an invariant map $s$ which induces an isometric cross section $\mathcal{Z}/G \to \mathcal{Z}_s \subset \mathcal{Z}$. Let $s \colon \mathcal{Z} \to \mathcal{Z}_s$ be defined as

$$s(z) = \sigma_z \cdot z, \tag{2}$$

where $\sigma_z \in S_n$ is a permutation which ensures that

$$z_{\sigma_z(1)} \leq z_{\sigma_z(2)} \leq ... \leq z_{\sigma_z(n)}. \tag{3}$$

In other words, $\sigma_z$ is the permutation that sorts the coordinates of $z$ in ascending order. This permutation clearly exists, since all sequences can indeed be sorted. While the sorting permuation $\sigma_z$ need not be unique, the sorted sequence will always be unambiguous.

The resulting map $s$ is clearly invariant, since for all $\sigma \in S_n$ we have that $\sigma z$ and $z$ will have the same form when sorted. Also, if we let $\mathcal{Z}_s = \{s(z) \mid z \in \mathcal{Z}\}$, then $s$ is surjective by definition. These observations, combined with Proposition 1, allow us to show the following results implying that $s$ does indeed induce an isometric cross section $s' \colon \mathcal{Z}/S_n \to \mathcal{Z}_s$:

**Proposition 2.** *Let $s \colon \mathcal{Z} \to \mathcal{Z}_s$ be the sorting function described above. Furthermore equip $\mathcal{Z}$ and $\mathcal{Z}_s$ with the Euclidean metric, and $\mathcal{Z}/S_n$ with the quotient metric. Then the induced $s' \colon \mathcal{Z}/S_n \to \mathcal{Z}_s$ defined as in Proposition 1 is a bijection as well as an isometry. Finally $\mathcal{Z}_s$ is a convex cone.*

The realization that $\mathcal{Z}_s$ is a convex is important, because it makes linear interpolation between elements of $\mathcal{Z}_s$ meaningful: Any point on the the line will be contained in the $\mathcal{Z}_s$ as well. A proof of Proposition 2 is included in the appendix.

## 3.2  WHEN NO ISOMETRIC CROSS SECTION EXISTS: RANDOM INVARIANT LINEAR PROJECTION

We cannot in general design an isometric cross section $s' \colon \mathcal{Z}/G \to \mathcal{Z}_s \subset \mathcal{Z}$, and for this more general situation we propose random invariant projections as a generic tool. Random projections are a well known alternative to trained dimensionality reduction techniques (Candes & Tao, 2006), and can be easily adapted to equivariant latent representations by using random invariant projections.

Random projections are often available: Ma et al. (2018); Maron et al. (2019) propose a basis for all permutation invariant linear maps, and (Cesa et al., 2022a) describe how $E(n)$-equivariant and invariant linear maps can be constructed from a basis. Initializing these layers at random, we obtain an analogy to the random projections known from classical statistics (Candes & Tao, 2006).

As we cannot generally invert random projection, these invariant representations do not allow interpolation-based analysis. However, they are still highly valuable for visualization and interpretation, as well as building new models and analyses directly from the invariant latent representation.

# 4 RELATION TO EXISTING INVARIANT, QUOTIENT AND EQUIVARIANT LATENT SPACES

Having presented our methods and notation, we can now explain in detail how they relate to recent related work. While the utility and interpretability of equivariant latent spaces is, to the best of our knowledge, unexpored in the past, the autoencoder literature explores latent space design. Mehr et al. (2018) design a quotient autoencoder for 3D shapes whose latents reside on the quotient space $\mathcal{X}/G$. Here, $\mathcal{X}$ parametrizes 3D shape and $G$ is the group of rotations or non-rigid deformations. As the latent space is $G$-invariant, shape alignment and interpolation are greatly simplified. However, as discussed above, quotient spaces are often non-Euclidean, greatly hindering their applicability.

Graph autoencoders commonly use a permutation *invariant* encoder. This can be achieved by using permutation equivariant layers (e.g. graph-convolutions) eventually followed by a permutation invariant layer Winter et al. (2021). As the composition of equivariant and invariant functions is invariant, this ensures that the latent representation is indeed invariant to permutation of the input nodes. This was the strategy followed e.g. in early graph VAE models Simonovsky & Komodakis (2018); Vignac & Frossard (2021); Rigoni et al. (2020); Liu et al. (2018), where an expensive graph alignment step was needed in order to train the model. To counteract this, newer models Hy & Kondor (2023) replace the invariant latent space with equivariant ones, similar to those studied in this paper. Similarly to the early VAE models, Winter et al. (2022) construct encoder-decoder architectures; here, however, the needed alignment of outputs with inputs is learned rather than optimized.

It is important to note that mathematically – ignoring implementation challenges – **all the above approaches are essentially equivalent**. If, as above, $h\colon \mathcal{X} \xrightarrow{f} \mathcal{Z} \xrightarrow{g} \mathcal{Y}$ is a predictor with equivariant latent feature embedding $f$, then the quotient map $\pi\colon \mathcal{Z} \to \mathcal{Z}/G$ composes with $f$ to form a quotient latent feature embedding $f_{quot} = \pi \circ f\colon \mathcal{X} \to \mathcal{Z}/G$ as was done in Mehr et al. (2018). The equivariant representation $\mathcal{Z}$ and the quotient latent representation $\mathcal{Z}/G$ carry *exactly* the same information, only with their own individual caveats – the quotient representation $\mathcal{Z}/G$ will often be non-Euclidean and cumbersome to work with, whereas – as we have seen – the equivariant representation $\mathcal{Z}$ does not have well defined representatives for each data point.

$$\mathcal{X} \xrightarrow{\phantom{aa}f\phantom{aa}} \mathcal{Z} \xrightarrow{\phantom{aa}g\phantom{aa}} \mathcal{Y}$$

More generally, any invariant latent feature embedding $f_{inv}\colon \mathcal{X} \to \mathcal{Z}_{inv}$ which carries enough information to enable decoding back onto $\mathcal{X}$, as in (Winter et al., 2022), can necessarily be written as a composition $f_{proj} \circ f_{quot}$ of a quotient latent feature embedding $f_{quot}\colon \mathcal{X} \to \mathcal{Z}/G$ for some latent space $\mathcal{Z}$, and a projection-like map $f_{proj}\colon \mathcal{Z}/G \to \mathcal{Z}_{inv}$. Thus, any differences in performance as observed in the experiments of Winter et al. (2022), are caused by implementation choices rather than differences in the underlying mathematical model – invariant, quotient and equivariant representations are, mathematically, able to carry the same information. We argue, that when taking care to respect the group action when utilizing and interpreting the equivariant latent representation $\mathcal{Z}$, there is no good reason to avoid it.

A final existing approach to obtaining invariant and equivariant maps is using fundamental domains Aslan et al. (2023). When isometric cross sections, they form fundamental domains. However, you cannot generally find isometric cross sections – indeed, the quotient space can exhibit strongly non-Euclidean geometry Calissano et al. (2023). This explains, in part, why the quotient itself is often not an efficient invariant representation. This also indicates that making an isometric, or even near-isometric, mapping to the representation space mathematically impossible. Random invariant projections, on the other hand, are in our cases designed as continuous mappings from the equivariant feature space, and they therefore preserve some local geometric structure.

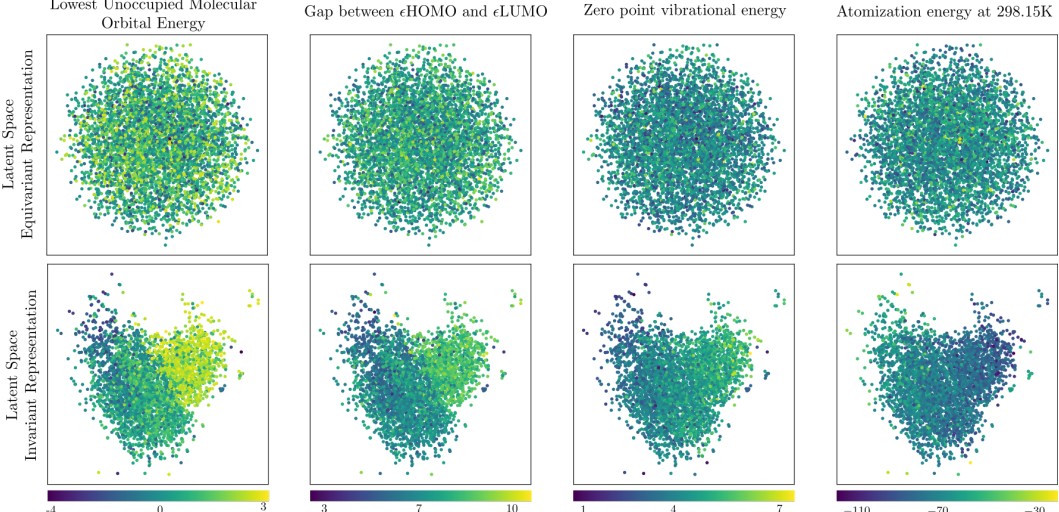

Figure 2: The first two principal components of the training data for the equivariant (top) and invariant representation (bottom). Each column illustrates a specific molecular property.

## 5 EXPERIMENTS

### 5.1 ISOMETRIC INVARIANT REPRESENTATIONS VIA A GRAPH VAE

We consider a permutation equivariant graph variational autoencoder (VAE) $h\colon \mathcal{X} \to \mathcal{Z} \to \mathcal{X}$ trained for molecule generation, and evaluate the utility of its latent codes for various downstream tasks. Each graph $(V, E) \in \mathcal{X}$ consists of a set $V$ of (at most) $n$ nodes with an $n \times d_A$ node feature matrix, and an $n \times n \times d_E$ edge feature tensor $E$, where $n$ denotes the number of nodes in the graph. A permutation $g \in S_n$ acts on the graph $(V, E)$ through its associated permutation matrix $P_g$:

$$g(V, E) = (P_g V, P_g E P_g^T). \tag{4}$$

The latent space of the VAE is designed as $\mathcal{Z} = \mathbb{R}^n$, and we let $s\colon \mathcal{Z} \to \mathcal{Z}_s$ be the invariant isometry defined by sorting function as defined in section 3.1.

**Dataset.** The QM9 dataset Ramakrishnan et al. (2014); Ruddigkeit et al. (2012) consists of approx. 130.000 stable, small molecules, using 80%/10%/10% for training/validation/testing. Each molecule is represented by at most 9 heavy atoms, their bindings (edges) and selected molecular properties. As our interest is in the permutation equivariant representation, we simplify the graphs to contain only the atom-type (node features) and binding-type (edge features). All graphs are padded with not-a-node and not-an-edge features to obtain the same node number for each graph.

**Model.** Our permutation equivariant VAE is based on linear equivariant layers as derived in Maron et al. (2018b), combined with entry-wise non-linearities, which were used in both the encoder and decoder. A comprehensive description of the model architecture can be found in appendix.

**Visualisation.** VAEs are often used to visualize disentangled latent representations (Mathieu et al., 2019; Mitton et al., 2021). Here we show how, when working with end-to-end permutation equivariant variational autoencoders, the obtained representations may by deceiving.

The top row of Fig. 2 shows the first two principal components of the equivariant latent representation[1] $\mathcal{Z}$ of the QM9 test set, with color representing different molecular properties. Inspecting the equivariant latent representations in the top row incorrectly suggests no apparent structure in the data, as molecules with similar molecular properties are by no means close in the latent space. On

---

[1]Note that the equivariant representations were obtained by randomly permuting the input nodes to remove any implicit ordering which may have been implied by the structure of the dataset.

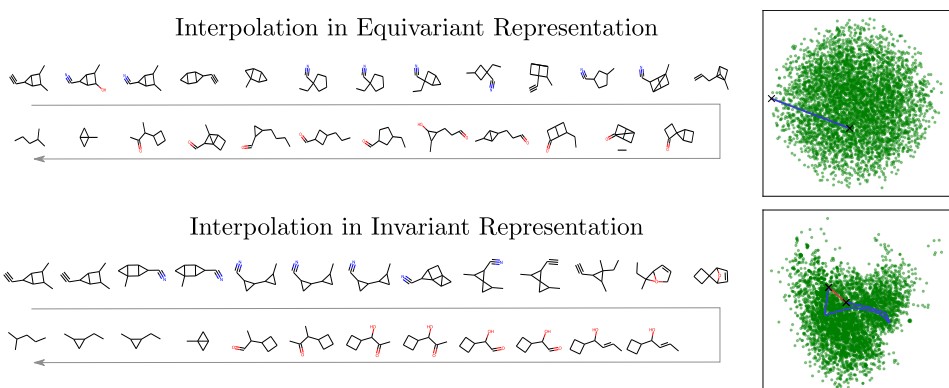

Figure 3: Molecules generated from interpolating between two molecules using the *equivariant* (top) and *invariant* (bottom) representations. Note that while the molecules decoded from $z_2$ and $s(z_2)$ differ in their embedding, they are equal up to permutation. Left: Molecules sampled along the two interpolations. Right: Interpolation in the latent space visualized via the first two principal components. In the equivariant representation, we visualize a straight blue line between $z_1$ and $z_2$. In the invariant representation, we visualize the linear interpolation between $s(z_1)$ and $s(z_2)$ (red), and the equivariant linear interpolation between $z_1$ and $z_2$ subsequently mapped to $\mathcal{Z}_s$ (blue).

the other hand, when inspecting the isometric invariant representation $\mathcal{Z}_s$ plotted in the bottom row, a different picture emerges. Here, we find a clear pattern between latent representation and molecular properties, indicating that the model does indeed pick up on important structure in the data. In other words, looking at *the same* latent representation using either the equivariant representation, or its invariant projection, leads to very different conclusions about both the data and the model.

**Latent space interpolation.** An autoencoder enables straightforward interpolation between molecules: Given two molecular graphs $\mathcal{G}_1, \mathcal{G}_2 \in \mathcal{X}$, we simply linearly interpolate between their respective latent representations and pass the resulting latent representation to the decoder, i.e.

$$\mathcal{G}_\alpha = g(\alpha f(\mathcal{G}_1) + (1 - \alpha) f(\mathcal{G}_2)), \quad \alpha \in [0, 1] \tag{5}$$

Here, we demonstrate how linear interpolation between latent codes $z_1$ and $z_2$ in the equivariant latent space $\mathcal{Z}$ may yield unstable decoded molecules, while linear interpolation between $s(z_1)$ and $s(z_2)$ in the isometric, invariant representation $\mathcal{Z}_2$ can remedy this issue. Note that linear interpolation between $s(z_1)$ and $s(z_2)$ is indeed meaningful, as we saw in section 3 that $\mathcal{Z}_s$ is convex.

Fig. 3 compares the equivariant interpolation $s(\alpha z_i + (1 - \alpha) z_j)$ to the isometric, invariant interpolation $\alpha s(z_i) + (1 - \alpha) s(z_j)$ by visualizing 25 decoded molecules sampled along each of the interpolating lines (left). The same two interpolating lines are compared through the lens of the isometric, invariant representation $\mathcal{Z}_s$ (right). We see that interpolation in the latent space yields pathological behavior, where molecule structure varies greatly along the line, whereas interpolation using the isometric, invariant cross section yields a far more smooth variation along the curve.

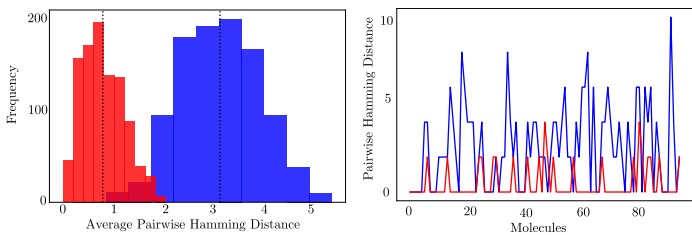

Figure 4: We investigate the pairwise Hamming distance between consecutive observation pairs sampled along the linear interpolation. Right: Histograms over the average pairwise distance for 10000 interpolations in the equivariant (blue) and invariant (red) representations. Left: visualization of the pairwise hamming distance along the single interpolation shown in Fig. 3.

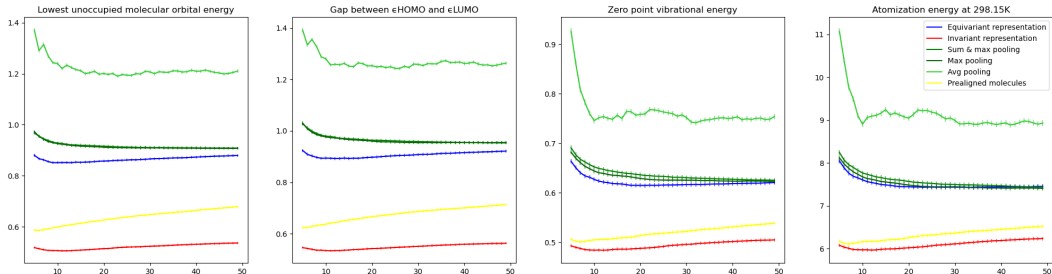

Figure 5: MAE vs $k$ for a kNN regression model predicting molecular properties from latent representations. We show MAE for the equivariant (blue) and isometric invariant representation (red), as well as invariant pooling operations commonly used to summarize equivariant representations.

We confirm this tendency of difference in interpolation smoothness by computing the pairwise Hamming distance between consecutive molecules sampled along the interpolating line. Fig. 4 shows quantitatively that consecutive graphs generated by interpolating in the equivariant representation, are less stable than those generated by interpolating in the invariant, isometric representation.

**Stability of molecules in a neighbourhood.** Using the test-set containing approximately 13.000 molecules, i.e. 10% of the available data, we now compare local structure of the equivariant- and invariant representations using a $k$-nearest neighbour regression model to predict molecular properties from latent representations. We report the mean absolute error (MAE), as is the custom for QM9 Wu et al. (2018), as a function of the number of neighbours considered, for each of the properties in figure 5. In addition to the isometric invariant representation, we also consider several invariant pooling operations commonly used to summarize equivariant representations. It is clear that the MAE is consistently lowest when training a predictor on the isometric invariant representations, indicating that they are better at preserving information needed for downstream analysis.

## 5.2 ROTATION INVARIANT MNIST CLASSIFIER

We demonstrate random invariant projections of equivariant latent features for a rotation invariant classifier trained on MNIST LeCun & Cortes (2010) augmented by rotation. Let $h\colon \mathcal{X} \xrightarrow{f} \mathcal{Z} \xrightarrow{k} \mathcal{Y}$ be a model where $\mathcal{X}$ is the space of images acted on by the rotation group $G = SO(2)$. Let $f$ be an equivariant feature embedding derived from the $E(n)$-equivariant architecture of (Cesa et al., 2022a), followed by an invariant pooling layer $k$

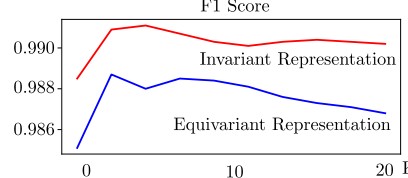

Figure 6: F1 score of KNN classifier.

and a fully connected classifier, making the architecture rotation invariant (see appendix for details).

The first two principal components of the equivariant and invariant representations are shown in Fig. 7. We see that naively using the equivariant representation yields a plot suggesting very little signal. However, after applying a random linear projection, the structure becomes strikingly clear. We analyse the local structure of the representations by applying a $k$-nearest neighbour classifier. Its quality is evaluated using the F1 score, see Fig. 6. Again, we see that the quality of the classifier increases when applied to the invariant representation as opposed to the equivariant one.

## 6 DISCUSSION

We have shown, for two commonly encountered types of transformations, how equivariant latent representations can lead to inappropriate and ambiguous interpretations, as they contain multiple representations per data point. Moreover, we have explained how equivariant latent representations implicitly encode well defined quotient representations. We show that for a particular permutation equivariant representation, this quotient representation $\mathcal{Z}$ admits an isometric cross section onto a well-defined subset $\mathcal{Z}_s$, retaining all information from the quotient $\mathcal{Z}/G$. More generally, we show how random invariant projections can produce informative invariant representations.

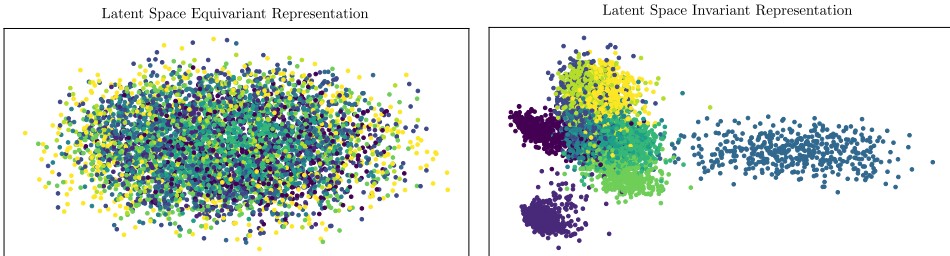

Figure 7: The figure shows the first two principal components of the training data for the equivariant representation (right), and the invariant representation (left). The colours represent the labels.

**Is the need for post hoc analysis a problem?**   Post hoc methods are sometimes considered inferior to intrinsic methods. We emphasize that the invariant representations presented here are comparable to dimensionality reduction methods such as UMAP, T-SNE or PCA – which are also post-hoc. The strength of our invariant representations is that they do not restrict the equivariant models themselves. This allows training models with performance only in mind, still obtaining invariant representations that are functions of the equivariant representations learned in model training.

**Weaknesses & future work**   While choosing an invariant mapping is a natural extension of the assumption of equivariant modeling, it may not always be clear how this map should be chosen, as this choice may depend highly on the model architecture. Suggesting suitable high-quality invariant mappings relevant for widely applied equivariant architectures is an obvious path for future work.

**Isn't this just an argument to avoid the mathematically cumbersome equivariant and invariant models?**   Invariant and equivariant models rely on advanced mathematical machinery which make them less accessible than standard deep learning models. One might be tempted to take our results as an argument to just avoid these models altogether. However, this would not suffice to avoid our illustrated problems. An alternative and common way to encourage invariance or equivariance is to rely on the inherent flexibility of deep learning models to learn approximate invariance/equivariance through data augmentation. Fig. 8 illustrates the effect of using augmentation to encourage rotation invariance in a classification CNN trained on MNIST Deng (2012) – we see a very similar pattern as in the previously described $SO(2)$ invariant classifier (right). On the left, we see how an intermediate latent representation of the CNN represents the MNIST test images when they are rotated randomly ("Augmented Dataset", left) or embedded using their original orientation ("Aligned Dataset", right). Since MNIST images have an orientation, they are naturally aligned with each other, and the resulting t-SNE plot nicely separates the different digits, obtaining an effect similar to the $SO(2)$ invariant classifier. For most image classification problems, however, there is no natural alignment, and a more realistic scenario is the plot on the left, where random rotations have been applied prior to embedding. What this shows is that while a pre-embedding alignment can work as a poor man's invariant representation, the lack of well-defined relative distances that we saw for strictly equivariant representations cannot be avoided by sticking to more straightforward models.

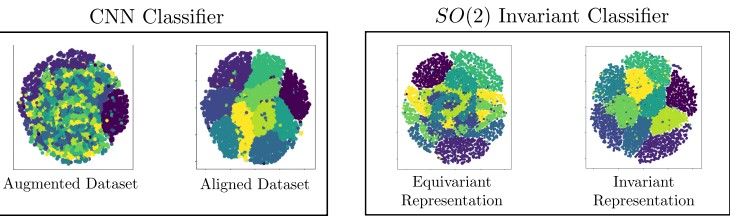

Figure 8: For the latent representation of a CNN classifier trained with rotation augmentation, rotated MNIST images (left) exhibit a similar embedding behavior as the equivariant representation of a $SO(2)$ invariant classifier, while the image alignment given by the natural orientation of digits provides a poor man's analogue of the invariant representation for the $SO(2)$ invariant classifier.

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

## A  APPENDIX

## B  PROOFS

This section will contain the proofs of central statements used in the paper.

### B.1  PROOF OF PROPOSITION 1

First we show that $s'$ is well defined. Pick $z_1, z_2 \in \mathcal{Z}$ and assume $[z_1] = [z_2]$. Then, by definition, $z_1 \in [z_2]$, and as a consequence, there exists some permutation $\sigma \in S_n$ such that $z_1 = \sigma z_2$. We now see that $s'([z_1]) = s(z_1) = s(\sigma z_2) = s(z_2) = s'([z_2])$, where the third equality follows from $s$ being invariant, and thus $s'$ is well defined.

We now show, that $s'$ is indeed surjective. Pick $y \in \mathcal{Z}_s$. Since $s$ is surjective, we have that there exists $z \in \mathcal{Z}$ such that $s(z) = y$. However, then $s'([z]) = s(z) = y$. Thus $s'$ is surjective.

### B.2  PROOF OF PROPOSITION 2

$s'$ **is bijective:** As a consequence of proposition 1, the function $s'$ is well defined, and surjective. It remains to show that $s'$ is injective. Pick $[z_1], [z_2] \in \mathcal{Z}/S_n$ and assume $[z_1] \neq [z_2]$. Then $[z_1] \cap [z_2] = \emptyset$, and thus there exists no permutation $\sigma \in S_n$ such that $\sigma z_1 = z_2$. Assume for contradiction that $s'([z_1]) = s'([z_2])$. Then we have that:

$$\sigma_{z_1} z_1 = s(z_1) = s'([z_1]) = s'([z_2]) = s(z_2) = \sigma_{z_2} z_2 \tag{6}$$

However, then $z_2 = (\sigma_{z_2}^{-1}\sigma_{z_1})z_1$. But as $S_n$ is a group $\pi_{z_2}^{-1}\pi_{z_1} \in S_n$, and thus we have a contradiction.

$s'$ **is an isometry:** We aim to show that $d([x], [y]) = d_{\mathcal{Z}_s}(s'([x], s'([y]))$. For all $[x], [y] \in \mathcal{Z}/S_n$, where $d$ denotes the quotient metric.

Pick $[x], [y] \in \mathcal{Z}/S_n$. Assume without loss of generality that $s(x) = x$ and $s(y) = y$, i.e. that the coordinates of $x$ and $y$ are already sorted. We can assume this as the coordinates of any representative can be sorted, thus achieving representatives with the properties we seek.

It now suffices to show that for any $\sigma \in S_n$:

$$d_{\mathcal{Z}_s}^2(x, y) = \sum_{i=1}^{n}(x_i - y_i)^2 \leq \sum_{i=1}^{n}(x_{\sigma(i)} - y_i)^2 = d_{\mathcal{Z}_s}^2(\sigma x, y) \tag{7}$$

If $\sigma$ is the identity permutation, then the above equation is trivially true. Also we note, that any permutation, can be written as a product of transpositions, thus if the above inequality is true for transpositions, then it will generalize to any permutation. Assume that $\sigma$ is a transposition, that is for $1 \leq j \leq k \leq n$ we have that $\sigma(j) = k$, $\sigma(k) = j$ and $\sigma(i) = i$ for $i \neq j, k$. Then

$$d_{\mathcal{Z}_s}^2(\sigma x, y) = \sum_{i=1}^{n}(x_{\sigma(i)} - y_i)^2 \tag{8}$$

$$= (x_k - y_j)^2 + (x_j - y_k)^2 + \sum_{i \in \{i | \pi(i) = i\}}^{n}(x_i - y_i)^2, \tag{9}$$

and therefore it now suffices to show that when $x_j \leq x_k$ and $y_j \leq y_k$ then

$$(x_j - y_j)^2 + (x_k - y_k)^2 \leq (x_k - y_j)^2 + (x_j - y_k)^2 \tag{10}$$

That this is indeed true, can be seen by observing that $x_k = x_j + c$ for some constant $c \geq 0$. Then

$$
\begin{aligned}
(x_j - y_j)^2 + (x_k - y_k)^2 &= x_j^2 + y_j^2 - 2x_jy_j + x_k^2 + y_k^2 - 2x_ky_k \\
&= x_j^2 + y_j^2 - 2(x_k - c)y_j + x_k^2 + y_k^2 - 2(x_j + c)y_k \\
&= (x_k - y_j)^2 + (x_j - y_k)^2 + 2c(y_j - y_k) \\
&\leq (x_k - y_j)^2 + (x_j - y_k)^2
\end{aligned}
$$

where the last inequality follows as $2c(y_j - y_k) \leq 0$ since $c \geq 0$ and $y_j \leq y_k$ by assumption. The assertion follows.

$\mathcal{Z}_s$ **is a convex cone:** We first show that $\mathcal{Z}_s$ is a cone, i.e. if $y \in \mathcal{Z}_s$ then $\alpha y \in \mathcal{Z}_s$ for all $\alpha \geq 0$. Assume $y \in \mathcal{Z}_s$. Then

$$y_1 \leq y_2 \leq ... \leq y_n \tag{11}$$

but then clearly

$$\alpha y_1 \leq \alpha y_2 \leq ... \leq \alpha y_n \tag{12}$$

which means that $\alpha y \in \mathcal{Z}_s$. Now, we show that $\mathcal{Z}_s$ is convex. Assume $x, y \in \mathcal{Z}_s$. Then:

$$x_1 \leq x_2 \leq ... \leq x_n \text{ and } y_1 \leq y_2 \leq ... \leq y_n \tag{13}$$

But then

$$x_1 + y_1 \leq x_2 + y_2 \leq ... \leq x_n + y_n \tag{14}$$

which implies that $x + y \in \mathcal{Z}_s$. Since $\mathcal{Z}_s$ is a cone it now follows that $\mathcal{Z}_s$ is also convex.

## C   PERMUTATION EQUIVARIANT VAE

By Maron et al. (2018a; 2019); Thiede et al. (2020); Pan & Kondor (2022) we have that we can define any linear function $L : \mathbb{R}^{n^k \times d} \to \mathbb{R}^{n^l \times d'}$ using exactly $b(k + l)dd'$ known basis elements, where $b(\cdot)$ denotes the Bell number. Using this result, we can define node- and edge-level linear equivariant layers:

$$L_V : \mathbb{R}^{n \times d_v} \to \mathbb{R}^{n^2 \times d'_e} \tag{15}$$

$$L_E : \mathbb{R}^{n^2 \times d_e} \to \mathbb{R}^{n^2 \times d'_e} \tag{16}$$

by using the a weighted linear combination of the known basis elements. That amounts to a total of $8d_v d'_v$ weights in the case of $L_V$ and $15d_e d'_e$ weights in the case of $L_E$. For an exact construction $L_V$ and $L_E$ please refer to Pan & Kondor (2022). Using this construction we can define a linear layer $L_{V,E} : \mathbb{R}^{n \times d_v} \times \mathbb{R}^{n^2 \times d_E} \to \mathbb{R}^{n^2 \times d'_E}$, by the channel-wise concatenation of $L(V)$ and $L(E)$. Note, that this concatenation does not change the equivariance property of the layer. All linear layers of the architecture utilised in the current work is of one of these forms. Subsequently a ReLU activation function is applied.

After each linear layer a 2D convolution with a $1 \times 1$ kernel size, a ReLU activation and instance normalization is applied. Again, all operations preserve the equivariance property of the network.

**Encoder:**  The encoder consists of a four linear layers:

- The first layer is a *hybrid* layer mapping the edge- and node-representation to a matrix representation similar to $L_{V,E}$.
- The two subsequent layers are equivariant linear layers mapping between matrix representations similar to the layers defined as $L_E$.
- The last layer, mapping to the latent representation, maps a matrix representation to a feature vector representation. In the current work we choose the number of latent channels to be 1.

Each of the linear layers, except for the last layer, are followed by 2D convolutions, ReLU activations and instance normalization as described above.

**Decoder:**  The decoder is likewise constructed using four linear layers:

- The first layer maps the latent feature vector to a matrix representation similar to $L_V$.
- The two subsequent layers map are equivariant linear layers mapping between matrix representations.
- The last layer is then the concatenation of a linear layer mapping between matrix representations (the reconstructed edge-matrix), and a linear layer mapping to a feature vector representation (the reconstructed node-matrix). The reconstructed edgematrix is enforced to be symmetric by adding the transpose of itself.

Again, following each layer except the last, we apply ReLU activations and instance normalization. In the last layer, a pointwise softmax is applied.

**Training details:**  The model was trained using the negative evidence lower bound (ELBO) as is standard for VAEs. A learning rate of 0.0001 and a batch-size of 32 was chosen. The model was trained for 1000 epochs. The QM9 dataset was obtained through the pytorch-geometric library Fey & Lenssen (2019).

## D  ROTATION INVARIANT MNIST-CLASSIFIER

The SO(2) invariant MNIST classifier is constructed using the tools provided in the ESCNN library provided by Weiler & Cesa (2019); Cesa et al. (2022b). That is, the model consists of 6 SO(2) steerable planar convolutions each of which is followed by batch-normalization and the FourierELU activation function. Each steerable planar convolution uses two irreducible representations to describe the output type; one invariant and one equivariant. The first two planar convolutions contains 16 feature maps, the next two 32 feature maps and the last two 64 feature maps. After each pair of planar convolutions a pooling layer is applied, and after the last convolution pooling is done over the spatial dimension to ensure invariance. Lastly, an invariant classifier is appended to the model. This is implemented using specifying a convolution using a kernel size of $1 \times 1$, and using the trivial representation to describe the output type, followed by a fully-connected classification network.

**Training details:**  The model was trained using a cross-entropy loss. A learning rate of 0.01 and a batch-size of 128 was chosen. The model was trained for 100 epochs. The MNIST dataset was obtained through the pytorch libraryPaszke et al. (2017).

