# OpenReview forum: "Interpreting Equivariant Representations"
_ICLR.cc/2024/Conference — Submitted to ICLR 2024_

### Official Review · Reviewer_t9rk · 2023-10-30

**Soundness:** 3 good
**Presentation:** 3 good
**Contribution:** 3 good
**Rating:** 6
**Confidence:** 4

**Summary:**

This paper identifies the problem of directly using equivariant latent representations for subsequent visualization tasks without considering the associated inductive bias of equivariance. The authors suggest employing an invariant projection of the latent representations in Euclidean space, enabling the application of conventional visualization tools to this transformed latent space.

To validate their approach, the authors present two illustrative examples. In the first scenario, they explore permutation equivariant autoencoders applied to graph data. Through this investigation, they find an isometric cross-section, ensuring the preservation of orbit distances in the space of the latent representation.

In the second scenario, the focus shifts to rotation-equivariant representations for the task of image classification. In this example, finding an isometric cross-section proves challenging. To address this, the authors propose to use random projections, showcasing their effectiveness through practical demonstrations.

**Strengths:**

- The motivation behind this work is solid. The majority of visualization tools operate in a Euclidean space, and the use of an equivariant representation is indeed problematic. This particular issue has not been extensively examined in existing literature, highlighting the novelty of this study.
- The paper is well-written. The writing strikes a balance between intuition and rigorousness. The presentation is well-paced and the figures for results are clear.
- The selected examples effectively substantiate the authors' assertions and are very interesting.

**Weaknesses:**

See questions.

**Questions:**

- Are there other significant examples where an isometric cross-section can be found? Or is the latent graph representation a very unique example? If that is the case, maybe the paper should focus more on the random projection approach because it has more applications.
- Is there any theoretical results to prove that random projections approximately find isometric cross-section? If this is not true, Sec 3.1 and Sec 3.2 feel like two completely unrelated methods.

---

> ### Author Response · Authors · 2023-11-16
> **Response to reviewer t9rk**
>
> Thank you for your time and your supportive feedback. Below, we aim to answer your questions – please ask again if you have further questions.
>
> **1. Are isometric cross-sections rare?**
>
> Unfortunately, we do believe that isometric cross sections are rare. However, permutation actions are commonly encountered, and we therefore find that the result warrants inclusion.
>
>
> **2. Can we prove that random projections are near isometric?**
>
> We don’t have any such results at present. We cannot expect to be able to prove full isometry: Quotient spaces such as Z/G are not generally Euclidean. Indeed, even for graphs with a permutation action, there is no guarantee that they even have bounded curvature. For such spaces, you cannot possibly obtain an isometric projection onto a linear representation, neither invariant nor otherwise. So isometry is more than you wish to ask for, and we don’t expect to be able to prove near isometry in general. However, for special cases, or locally, it might be feasible. We will do our best to clarify this in the paper.
>
> We agree that the two invariant visualization techniques are rather different, but we also find that they are tied together by attempting to solve the same problem – which is not generally well known either.

---

> > ### Comment · Reviewer_t9rk · 2023-11-22
> >
> > I have read the authors' answer and will keep my original rating. Thanks for explanation.

---

### Official Review · Reviewer_Nrx9 · 2023-11-01

**Soundness:** 3 good
**Presentation:** 4 excellent
**Contribution:** 2 fair
**Rating:** 5
**Confidence:** 2

**Summary:**

This paper proposes that latent equivariant representations should not be used naively and that invariant representations should be used. Equivariant representations should be projected in an invariant manner. The authors show empirically on a permutation equivariant graph VAE and a SO(2) invariant classifier on MNIST that invariant representations are more interpretable.

**Strengths:**

The authors provide a clear background to equivariant and invariant representations.

Figure 3 helps visualize the interpolation in latent space and helps the reader analyze the interpretability.

**Weaknesses:**

The fundamental argument of the paper seems to be that existing visualization methods cannot handle quotient spaces correctly. If I remember correctly, doesn’t UMAP map the data to a lower-dimensional manifold embedded in higher-dimensional space? Why would UMAP not be able to handle quotient spaces?

It’s unclear to me that invariant representations would be more interpretable than equivariant ones. Since equivariant representations inherently generalize invariant representations, why would invariant representations increase interpretability?

**Questions:**

See weaknesses.

Typo: Abstract Line 4: “inductive bias imposed on the by an equivariant”

---

> ### Author Response · Authors · 2023-11-16
> **Responses to reviewer Nrx9**
>
> Thank you for your time and for your feedback on our paper. Below, we seek to answer your questions – please feel free to let us know should you have further questions
>
> **1. Why can’t we just use U-map?**
>
> The U-map [1] considers situations where data measured as vectors (thus residing in a Euclidean space R^n) actually reside on a Riemannian submanifold M of R^n. U-map tries to unfold this manifold M with a projection of the data onto it.
>
> Our situation is actually different. The equivariant representation X is Euclidean, but because of the equivariance, each data point has a large number of valid representatives in X, and these could be measured as scrambled all over X – the assumption that these randomly scrambled representatives would reside on a submanifold M is clearly invalid. This is also illustrated by Figure 1 and 2 in our paper.
>
> The data points are, however, uniquely represented on the quotient Z – but Z is not generally Euclidean. It is also not generally a manifold at all, and there is thus no guarantee that the data resides on a manifold.
>
> To conclude, U-map is not applicable to our case. Notice that you could use U-map on top of the invariant representation thus making U-map invariant as well.
>
>
> **2. Why are invariant representations more interpretable?**
>
> Invariant representations increase interpretability because they only carry one single representative of each data point. For the equivariant representation of N generic graphs with n nodes each, you will have (n!)^N! different embeddings, with different corresponding plots, visualizing the exact same representation of the exact same data, with pairwise distances that vary significantly and discretely from plot to plot. As a direct consequence of this, when using a non-invariant dimensionality reduction methods such as for example PCA, we obtain visualizations (see the first row of Figure 2), which are indeed very hard to interpret, and may lead us to wrong conclusions, e.g. that the model did not train properly, or that it did not pick up on any structure in our data.
>
> The invariant representation, on the other hand, is unique, and only continuously deforms the distances from the quotient, similar to what we are used to in dimensionality reduction tools.
>
>
> [1] McInnes, L., Healy, J., & Melville, J. (2018). Umap: Uniform manifold approximation and projection for dimension reduction. arXiv preprint arXiv:1802.03426.

---

> > ### Comment · Reviewer_Nrx9 · 2023-11-23
> >
> > I've read the authors's response and thank you for clarifying. However, I'm still unconvinced of the significance of the work. It's not clear to me that equivariant representations can't also represent invariant quantities e.g. in irreducible representations. I therefore maintain my original score.

---

### Official Review · Reviewer_f5He · 2023-11-01

**Soundness:** 2 fair
**Presentation:** 1 poor
**Contribution:** 2 fair
**Rating:** 5
**Confidence:** 2

**Summary:**

This paper argues that using equivariant representations for downstream analysis can be misleading since they can assign distinct representations to inputs that only differ by some group action. To fix this they propose projecting the equivariant representations with an invariant projection which then ensures that the projected representation preserves invariances. There are however challenges in designing such an invariant projection since it must also retain information. This paper proposes such transformations and shows that they can lead to more reliable interpretation of representations. The efficacy of the proposed method is shown on a molecule graph generation VAE and an MNIST classifier.

**Strengths:**

1. The paper thoroughly explains why distances in an equivariant representation space can be misleading
2. The paper explains why an invariant transform is the right way to analyze equivariant representations
3. The paper also explores different possible invariant transforms that can better analyze representations while retaining information
4. The theory is corroborated with experimental results

**Weaknesses:**

1. I had a hard time reading and contextualizing this paper. For example, I'm not fully sure I understand why one would use equivariant representations when the inputs do not follow the G-equivariance of the representations. For eg, if x and g.x are similar inputs, then an equivariant representation is perhaps not the correct one to analyze such data in the first place. Why wouldn't one, instead just choose some mapping f that is invariant?

2. I did not fully understand Section 4, it read more like a related work section and felt out of place.

3. It was never clear to me what "analysis of representation" meant throughout the paper. All the results indicate that "analysis" means visualization of the representations in a 2D space. Is that the only "analysis" where equivariant representations would show deceiving results? It would help with readability to clarify what kinds of analyses are hard with equivariant representations.

4. Isn't the interpolation experiment self-satisfying? You already claim that after the invariant transform the representation space is convex and thus interpolation makes total sense (and thus the nicer results in Fig 4), whereas this is not the case for equivariant representations (thus results are all over for these in Fig 4). I'm not quite sure what was the purpose of this experiment. Was it to empirically validate the theory?

5. Overall, the paper was a tough read for me. It would be very helpful to start by contextualizing a use case where equivariant representations might need a downstream analysis where equivariance can be misleading, this would help motivate the paper better.

**Questions:**

See weaknesses.

---

> ### Author Response · Authors · 2023-11-16
> **Responses to reviewer f5He**
>
> Thank you for your time and efforts reading our paper, and for your constructive feedback. Below, we will try to respond to your questions and concerns.
>
> **1. Why don’t you just use an invariant mapping when x and g.x are similar?**
>
> We don’t quite understand the reviewer’s comment – we don’t study cases where the inputs x and g.x are similar, we study cases where they are different – e.g. permuted graphs or rotated images. In such applications, when your model predicts a structured output (e.g. a reconstructed graph, or an image segmentation) you want your model to be equivariant, not invariant.
>
> You are right, however, that invariant models are also applied to this type of data – the difference consists of whether or not the output should be transformed analogously to the input (as when the output is a segmentation or a reconstruction), or whether the output does not change when the input is transformed (e.g. when the output is a class). When building classification networks, the standard approach is to stack equivariant layers and then end the model with an invariant layer thus obtaining a fully invariant model.
>
> **2. Isn’t Section 4 a related work section?**
>
> Section 4 is indeed a related work section. We positioned this section here to aid the reader, as the mathematical framework introduced in the previous sections is needed to  fully explain how our methods relate to previously published work. Thank you for pointing out that this was confusing; we have added a comment to make this more clear – please see the updated paper.
>
> **3. Why would you analyze an intermediate equivariant representation?**
>
> Latent codes are widely used for downstream tasks, examples of which have been included below. The key observation here, is that not all these methods can be designed to be invariant, and as such the obtained results may be deceiving. In the paper we therefore include a range of very different examples where the non-invariance of the applied methods poses a problem (i.e. when doing PCA, interpolation or naive classification based on the latent codes). As these methods are very different in nature when applied to the latent codes, we use the phrase “analysis of the representation” to refer to any non-invariant method which could naively be applied directly to the equivariant representation.
>
> Important examples how equivariant latent representations are used:
>
> - Predictive models (classification, regression) where the intermediate representation is learned with unsupervised or self-supervised pretraining. Classically, the VAEs studied in our papers have been popular for this task; at present masked autoencoders are gaining popularity. In all of these, you predict objects with similar structure (and similar symmetries) as the input. Thus, you would naturally require or encourage (through augmentation) the model to be equivariant. We provide such examples in the paper, in Figures 5 (graph property regression) and 6 (image classification).
> - Generation/discovery of new objects such as molecules, proteins, or other structures, with sought properties – such analysis is often based on VAEs, which would typically be required to be equivariant. This relates to the interpolation experiment shown in Fig. 3, where interpolation would typically be used to encourage discovery of new molecules that combine the properties of the two points being interpolated.
>
> **4. Isn’t the interpolation experiment self satisfying?**
>
> The equivariant representation X is convex. However, when performing convex interpolation, it might suffer from the problem of potentially running into non-aligned representatives of the interpolated graphs, images, etc. The cross section Z_s does not have this problem because it only contains one representative of every graph, and these are geometrically aligned (because Z_s is isometric to the quotient). The fact that the cross section Z_s is a convex subset of  the equivariant representation X (see Proposition 2) makes the interpolation computationally and mathematically very easy. The purpose of the interpolation experiment is to empirically demonstrate that this makes a real (and not just mathematical) difference. In our experience, this is important for conveying a message to the more empirical parts of the community.
>
> **5. Starting by contextualizing the paper would help readability**
>
> Thank you for pointing this out. We have adapted the introduction to emphasize the use of autoencoders for learning self- and semi-supervised embeddings as a prominent use case. Note that equivariant representations are either enforced or encouraged (via augmentation) in any autoencoder, including the VAEs and masked autoencoders commonly used for semisupervised and self supervised learning. These representations are, e.g., typically used for downstream classification, discovery of novel structures like proteins or molecules, but also for specialized tasks such as counterfactual explanations or graph/image editing.

---

> > ### Comment · Reviewer_f5He · 2023-11-23
> > **Thanks for the response!**
> >
> > I appreciate the authors for taking the time to respond! Many things are more clear now.
> >
> > I'd like to clarify that re #3 ("Why would you analyze an intermediate equivariant representation?") -- what I meant in my original review was that it's not clear what **kind of analyses** are of interest here (I definitely understand the more general need to analyze representations). Nonetheless, I appreciate the response and it's clear now that you meant analysis in the more general sense of "using the representation for some downstream task". Although the broader point (which ties into what I wrote in #1) still stands, when the downstream task you care about (let's say classification) requires x and g.x to be the same (say x is an image and g is a small rotation) you want both x and g.x to be classified as the same and hence by definition you want invariance. So using an equivariant representation, just by definition, seems wrong here (unless, as you mentioned, you stack some invariant layers on top of this equivariant representation, which is what is done eg in many popular CNNs). You mentioned in your response when referring to masked autoencoders: "Thus, you would naturally require or encourage (through augmentation) the model to be equivariant", however, at least for self-supervised vision models (including masked autoencoders), *invariance* is encouraged by making sure two views of the same input are mapped to the *same* representation (eg, SimCLR[1], SimSam[2]). At least to me, it's not obvious why equivariant representations are relevant for such tasks, and thus I (still) fail to fully understand the motivation for this work.
> >
> > I've also read other reviewer's comments and would keep my score, but it's possible that I've misunderstood the point of the paper and won't stand in the way of acceptance if others feel strongly about it.
> >
> > [1] https://arxiv.org/abs/2002.05709
> > [2] https://arxiv.org/abs/2011.10566

---

### Meta-Review · Area_Chair_4sbu · 2023-12-07

**Metareview:**

This paper demonstrates that when using equivariant representations for downstream tasks, the downstream model should respect this equivariance otherwise face with lower performance. They propose to use invariant projection to accomplish this. Consistent with two fo the reviewers, I find the paper difficult to follow at times and the motivation/studied setting not very clear. Furthermore, the empirical results also ins't very convincing in demonstrating the use case of the approach. The AC encourages the authors to better structure and motivate the problem they are solving; For example, a running example or illustration of the problem would benefit the presentation.

**Justification For Why Not Higher Score:**

The presentation of the paper is not ready for publication. Especially, the motivation of the work needs to be strengthened to provide a better sense on why the studied question is worth studying.

**Justification For Why Not Lower Score:**

N/A

---

### Decision · Program_Chairs · 2024-01-16

Reject